# On the reproducibility of
# "Exacerbating Algorithmic Bias through Fairness Attacks"

## Reproducibility Summary

**Scope of Reproducibility**

The paper presents two novel kinds of adversarial attacks against fairness: the IAF attack and the anchoring attacks. Our goal is to reproduce the five main claims of the paper. The first claim states that using the novel IAF attack we can directly control the trade-off between the test error and fairness bias metrics when attacking. Claims two to five suggest a superior performance of the novel IAF and anchoring attacks over the two baseline models. We also extend the work of the authors by implementing a different stopping method, which changes the effectiveness of some attacks.

**Methodology**

To reproduce the results, we use the open-source implementation provided by the authors as the main resource, although many modifications were necessary. Additionally, we implement the two baseline attacks which we compare to the novel proposed attacks. Since the assumed classifier model is a support vector machine, it is not computationally expensive to train. Therefore, we used a modern local machine and performed all of the attacks on the CPU.

**Results**

Due to many missing implementation details, it is not possible to reproduce the original results using the paper alone. However, in a specific setting motivated by the authors' code (more details in section 3), we managed to obtain results that support 3 out of 5 claims. Even though the IAF and anchoring attacks outperform the baselines in certain scenarios, our findings suggest that the superiority of the proposed attacks is not as strong as presented in the original paper.

**What was easy**

The novel attacks proposed in the paper are presented intuitively, so even with the lack of background in topics such as fairness, we managed to easily grasp the core ideas of the paper.

**What was difficult**

The reproduction of the results requires much more details than presented in the paper. Thus, we were forced to make many educated guesses regarding classifier details, defense mechanisms, and many hyperparameters. The authors also provide an open-source implementation of the code, but the code uses outdated dependencies and has many implementation faults, which made it hard to use as given.

**Communication with original authors**

Contact was made with the authors on two occasions. First, we asked for some clarifications regarding the provided environment. They promptly replied with lengthy answers, which allowed us to correctly run their code. Then, we requested additional details concerning the pre-processing of the datasets. The authors pointed at some of their previous projects, where we could find further information on the processing pipeline.

---

# 1 Introduction

Machine Learning models have shown impressive performance in countless domains in the last decade. However, it has been demonstrated that an adversary can input carefully-crafted perturbations to subvert the predictions of these models. The area of Adversarial Machine Learning has emerged to study vulnerabilities of machine learning approaches in adversarial settings and to develop techniques that make them robust against malicious attacks.

Most of the research has focused on studying malign interventions that degrade the accuracy of a system: imagine, for example, the consequences of inducing wrong predictions in an autonomous driving system. Only recently, fairness has become a rising concern for the performance of machine learning models, especially for sensitive fields such as criminal justice and loan decisions. Along these lines, "Exacerbating Algorithmic Bias through Fairness Attacks" [1] proposes two families of poisoning attacks that inject malicious points into the models' training sets and intentionally target the fairness of a classification model.

The first, the *influence* attack, extends the optimization-based technique introduced by Koh et al. [2] by incorporating in the loss function a constraint for fair classification. An attacker can hence harm both accuracy and fairness simultaneously, with a trade-off regularized via a parameter $\lambda$. The second type of attack, the *anchoring* attack, affects solely fairness and aims to place poisoned data points to bias the decision boundary without modifying the attacker loss. Depending on whether the target point is chosen at random, anchoring attacks are classified as *random* or *non-random*.

# 2 Scope of reproducibility

This report investigates the reproducibility of the original paper by Mehrabi et al. and aims to verify its main claims. Since these heavily rely on the datasets and metrics used by the authors, the reader is invited to consult Sections 3.2 and 3.3 – respectively – for a refresh of such concepts. Then, the main claims can be summarized as follows:

– *Influence* **Attack on Fairness (IAF)**:

- *Claim 1:* Increasing the parameter $\lambda$ results in stronger attacks against fairness. Contrarily, for lower values the model acts similarly to the original influence attack [2] targeted towards accuracy;
- *Claim 2:* The proposed IAF outperforms the attack of Koh et al. [2] in affecting both fairness metrics (SPD and EOD), on all three datasets;
- *Claim 3:* The proposed IAF also outperforms the attack based on the loss function proposed by Solans et al. [3] in affecting SPD and EOD, on all tested datasets.

– *Anchoring* **Attack**:

- *Claim 4:* Both random and non-random anchoring attacks (RAA and NRAA, respectively) outperform Koh et al. [2] in degrading the SPD and EOD of the classification model, on all three datasets;
- *Claim 5:* On the German and Drug Consumption datasets, RNAA and NRAA have a greater impact on fairness metrics (SPD and EOD) compared to the attack based on Solans et al. [3]. However, the latter outperforms the proposed anchoring attack in affecting fairness when classification is performed on the COMPAS dataset.

# 3 Methodology

The authors provided an open-source implementation of their code on GitHub [4]. Unfortunately, the repository has several issues: dependencies are not sufficiently specified, and simply running the code in the given environment results in conflicts. Furthermore, the code does not provide an option to run baseline methods used in the paper, nor does it include the essential hyperparameter $\lambda$, which is used in the experiments. The majority of the code is based on Koh et al. [2]'s public implementation [5], and a code coverage analysis revealed that more than 50% is not used for running experiments related to this paper[1]. Moreover, the repository comes with pre-processed datasets and while this may sound advantageous, there is no mention of the processing procedure in the paper nor on GitHub. Finally, the code is generally complex and hard to understand due to insufficient comments and documentation.

Therefore, we used the codebase provided by the authors and customized it for our purposes. First, to aid maintainability and scalability, as well as to ensure future reproducibility of the original experiments, the code was modernized and made compatible with the latest version of *every* dependency. This involved major changes to migrate from `Tensorflow`

---

[1]The `coverage.py` tool [6] was used to measure code coverage, and the study was performed considering all possible attacks-datasets combinations.

1.12.0 to 2.6.2 and to update CVXpy from version 0.4.11 to 1.1.18[2]. Secondly, datasets were downloaded from the original sources [7, 8] and processed from scratch. The procedure is thoroughly reported in Section 3.2. Furthermore, the code was trimmed down to the essential, and the user was given the option to choose any of the available models and the corresponding parameters. Lastly, we added comprehensive documentation to make the code more interpretable.

## 3.1 Model descriptions

It appears that the authors of the original paper do not specify the model that they use for the given classification task. From the implementation details given in Koh et al. [2], as well as from [1]'s codebase, we assume the use of a Support Vector Machine (SVM) trained with a smooth hinge loss and L2 regularization (refer to [2] for further details). Additionally, the optimization algorithm is not indicated; we assumed it to be Newtons Conjugate Gradient (Newton-CG) method, as suggested by the codebase. Such a method is used for both the minimization of the parameters on the training set and the update step of the poisoned points (for attacks utilizing an adversarial loss). The gradient is computed using the full datasets, i.e., without using mini-batches. Although hardly recognizable, this follows the implementation of the original paper: from our interpretation of the code, it seems that the authors define a variable containing the size of the mini-batch size and the necessary functionality, but then never use it.

Our base algorithmic setup for the IAF, RAA, and NRAA attacks is described in the *Methods* section of the original paper. However, the authors omitted important details that we consequently had to assume based on more or less concrete evidence. First, an advantaged and disadvantaged group for the sensitive attribute (i.e., gender, as per the original work) has to be specified for all attacks. Since the rationale behind this choice does not seem to be included in the paper, we infer from the codebase that the authors did it automatically and deduced it from the datasets. More specifically, we assume that the advantaged group is chosen as the group with the highest ratio of data points with positive label ($y = 1$), regardless of the actual class label it corresponds to. This method is simple yet fallacious: for instance, it means that the group taking on the label "likely to perform a crime soon" more often (in the context of the COMPAS dataset) is considered "advantaged" in terms of the algorithm.

Secondly, for the computation of the feasible set using an anomaly detector $B$, we assume that the intersection of the Slab defense and the L2 defense was originally employed, as described in Koh et al. [2]. For reprojecting poisoned data points into the feasible set, we again use the approach of [2], which incorporates LP rounding for discrete variables.

Moreover, we implement two baselines. The three proposed attacks are compared against the original accuracy-targeting attack proposed by Koh et al. [2], and another attack that uses a loss function proposed by Solans et al. [3], which targets fairness[3]. Lastly, the model-specific changes/improvements are presented below:

**IAF.** As mentioned before, we modified the code to include the hyperparameter $\lambda$ which controls the trade-off between the accuracy and the fairness loss in the adversarial loss.

**Koh attack.** We were not able to find a way of running this baseline attack using the given codebase. We have decided to implement it from scratch, treating it as the limiting case of the IAF attack when $\lambda = 0$ (meaning no fairness loss in the adversarial loss function). Consequently, it is not exactly as presented in [2]: in the original Koh attack sampling, the initial poisoned points are not drawn from advantaged and disadvantaged groups, contrary to the IAF attack. However, we argue that equalizing the sampling method provides a stronger comparison between the two methods, as we alleviate the issue of the missing inductive bias from the original Koh influence attack.

**Solans attack.** This attack serves as the second baseline. We could not find it in the codebase, thus we implemented it by replacing the adversarial loss in the IAF attack with a weighted sum loss, as presented in [3]. Implementing this change posed a bigger issue than expected, due to the inflexibility of the TensorFlow-based implementation. Thus, major revisions were required.

## 3.2 Datasets

The authors provide compressed npz files of the three real-world datasets used for their experiments – the German Credit Dataset [7], the COMPAS Dataset [8] and the Drug Consumption Dataset [7]. However, these are already pre-processed, and the processing procedure is not reported nor documented in the code. This constitutes an important reproducibility barrier, because raw datasets[4] are not directly usable with the given codebase.

---

[2]In our repository we provide a YAML configuration file to quickly set up the required environment.

[3]For simplicity, we will refer to the influence attack presented in [2] as the Koh attack, and we will also refer to the attack presented in [3] as the Solans attack.

[4]The German Credit Dataset and the Drug Consumption Dataset can be downloaded from the UCI machine learning repository [7], while the COMPAS can be found in the corresponding GitHub repository[8].

In this section, we present our pre-processing pipeline, which was mainly determined by reverse engineering of the given files. Like the authors, we provide a set of `npz` files containing already-processed data to run our implementation, but we also include the scripts used to pre-process each dataset in the `Custom_data_preprocessing` directory. Lastly, to run the attacks, we assume that the advantaged and disadvantaged groups are males and females respectively. We accordingly map them to 0 and 1 to create the `group_label` binary array.

In the rest of this section, we outline our dataset-specific details of the pre-processing pipeline and the assumptions that were made for the sake of reproducibility of the original results.

**German Credit Dataset.** The dataset contains the credit profile of 1000 individuals with 20 attributes associated with each person. In our experiments, we use all of them, as in [1]. The attributes are both numerical and categorical, and we assumed the original authors used *one-hot* representations to encode the latter. The assumption was based on an extensive study of the provided datasets, with particular attention to their shapes. We then autonomously standardize the data, as it is common practice in Machine Learning, and split the data into an 80-20 train and test split, as indicated in the original paper.

**COMPAS Dataset.** ProPublica's COMPAS dataset [8] contains information about 7214 defendants from Broward County. We use the features specified in Table 1 of [1]. In this case, based on the provided dataset, we concluded that the authors must have used *numerical label* encoding to represent the categorical attributes. Finally, we standardize the data and split it into an 80-20 train and test split.

**Drug Consumption Dataset.** The dataset contains information about the drug consumption of 1885 individuals [9]. We use the attributes indicated in Table 1 of the original paper. The pre-processing procedure is as follows: first, we binarize the categorical data linked to cocaine consumption into *users* and *non-users*. Intuitively, non-users should be mapped to 0 (and 1 in the opposite case), but an inspection of the provided `npz` file suggests that the authors reversed the mapping. We decided to adhere to their choice for the sake of reproducibility. Moreover, we suspect that the dataset was shuffled before splitting it into training and test sets[5]. By doing so, we obtain similar results in the experiments. Finally, we standardize the data. The original processing of this dataset was particularly difficult to replicate, because contrary to what was reported in the paper, the authors did not follow an exact 80-20 train and test split. Rather, the two contained 1500 and 385 data points respectively.

To conclude, it is noteworthy that even the pre-processed datasets provided by the authors are not immediately usable: the position (specified as index) of the sensitive feature (i.e., gender) is different for each dataset and is only given for the German dataset in the running instructions. To account for this unnecessary confusion, our custom pre-processing procedure includes the moving of the gender column to the $0^{th}$ index, which is taken as default by the main function. In this way, we simplify the running instructions and make them coherent across datasets. Still, the user is given the ability to pass the sensitive feature index as an argument, to facilitate future experiments on different and untested data.

## 3.3 Metrics

The attacks are evaluated in terms of accuracy and fairness. Along with classification (test) error, the original paper uses two important metrics to evaluate the attack in terms of fairness: Statistical Parity Difference and Equality of Opportunity Difference.

**Statistical Parity Difference.** Statistical Parity Difference (SPD) was first introduced by Dwork et al. [11] and is used to capture the predictive outcome differences between different (advantaged and disadvantaged) demographic groups. The mathematical formulation is reported in Equation 1.

$$SPD = \left| p\left(\hat{Y} = +1 \mid x \in \mathcal{D}_a\right) - p\left(\hat{Y} = +1 \mid x \in \mathcal{D}_d\right) \right| \tag{1}$$

where $\mathcal{D}_a$ denotes the advantageous group and $\mathcal{D}_d$ denotes the disadvantageous group.

**Equality of Opportunity Difference.** Equality of Opportunity Difference (EOD) (Hardt et al. [12]) captures differences in the true positive rate between different (advantaged and disadvantaged) demographic groups. It is defined as shown in Equation 2.

$$EOD = \left| p\left(\hat{Y} = +1 \mid x \in \mathcal{D}_a, Y = +1\right) - p\left(\hat{Y} = +1 \mid x \in \mathcal{D}_d, Y = +1\right) \right| \tag{2}$$

---

[5]The main author followed a similar pre-processing procedure in another project that is publicly available on their GitHub [10].

### 3.4 Experimental setup and hyperparameters

All experiments shown in this paper can easily be reproduced using our code, which is publicly available on GitHub[6]. There we also provide technical details on how to run experiments and test different attacks in various settings. In this section, however, we list some additional details necessary to replicate the exact setup.

- The original code constrains the maximum iterations of an attack to 10000 and uses early stopping to interrupt training if the accuracy on the test set does not decrease for a specific number of iterations, which is hardcoded to be 2. We follow this strategy but adapt it for our experiments. First, we implement early stopping on both accuracy and fairness, meaning that the user can also choose to stop training in the absence of changes in fairness. We utilize *average fairness* $(SPD + EOD)/2$ as the stopping criteria[7] since the two metrics have similar behavior and equal range $[0, 1]$. Then, we set the early stopping patience as a controllable hyperparameter.
- It is unclear from the paper how the best-performing model was selected by the authors. The code suggests the usage of the model after the last attack iteration and training of the model parameters. Instead, we decided to save the best-performing model on the test set according to the chosen stopping metric (average fairness or accuracy), to better reflect the actual best performance. By selecting the best model based on fairness, we hope to choose more relevant states of the poisoned data affecting the fairness metrics. We compare the results in Section 4.
- The computation of the feasible set and the reprojection of poisoned points onto it is handled as a convex optimization problem (see [2]). Since we upgraded `CVXpy` to its newest version, we can let the library select the most appropriate solver for the given problem, instead of specifying one (the authors of [1] seem to have used the SCS solver).
- Following the original implementation, we utilize the `fmin_ncg` optimizer of the `scipy` library [13] for the Newton-CG optimization. We comply with the choices of the authors and set the convergence threshold of the fmin optimizer to $10^{-8}$, and the maximum number of iterations to 100. We follow the implementation details specified in [2] for computing the inverse Hessian-vector.
- During training, the temperature of the smooth hinge loss is chosen to be $0.001$, as found hardcoded in the original implementation. The value for the weight decay is set to $0.09$ for all datasets (apart from the code of the authors, this assumption is also backed up by the main experiments of Koh et al. [2]). The step size utilized in the IAF algorithm (and thus also in the Koh and Solans attack) is set to $0.1$ for all experiments, as found in the codebase.
- The threshold of the anomaly detector (see [2]) is controlled by a hyperparameter named `"percentile"`, which specifies the percentage of the data left after applying the anomaly detector. We first experimented with a value of 95 as suggested by Koh et al. [2] but, as this seemed to lead to some training failings, we settled on 90 (the default value given in the codebase).
- The number of injected poisoned points is proportional to the number of clean data points, such that $|\mathcal{D}_p| = \epsilon |\mathcal{D}_p|$ (where $\mathcal{D}_c$ and $\mathcal{D}_p$ are the set of clean and poisoned data points respectively). The authors control such quantity by using the proportionality factor $\epsilon$ as a changeable parameter. Accordingly, we do the same and also make $\lambda$ a controllable parameter.
- After careful inspection and testing of the authors' code, the EOD metric calculation was found to be faulty and was consequently re-implemented. Our adaptation is based on the paper that originally proposed it [12] and inspired by the implementation found in the `AIF360` library [14].
- Finally, the distance to original points in anchoring attacks $\tau$ was set to 0 for all experiments, as in the original paper.
- The random seed in all experiments was set to 1

### 3.5 Computational requirements

To give a complete overview of our experimental setup, we collect the average runtimes per iteration for different datasets and types of attacks. These are presented in Table 1. All models have been trained on a local machine with an AMD Ryzen 5 5600x CPU (6 cores, Base clock 3.7 GHz). Since the datasets are small, there is no need for more than 4Gb of RAM. In this sense, training should be virtually possible on any entry-level PC.

## 4 Results

### 4.1 Results reproducing original paper

As stated in Section 2, five main claims were identified in the original paper. In our specific setting, we were able to reproduce three of these, as summarized in Table 2. In this section we elaborate on our reproduction results: first, in

---

[6]https://anonymous.4open.science/r/MLRC2021_fairness_attack/

[7]In the rest of the paper, we might refer to it simply as the *fairness* stopping metric.

| Attack | German dataset [s] | COMPAS dataset [s] | Drug dataset [s] |
|--------|--------------------|--------------------|--------------------|
| IAF    | 0.870              | 0.265              | 0.312              |
| NRAA   | 1.123              | 106.23             | 3.678              |
| RAA    | 0.934              | 0.306              | 0.324              |
| Koh    | 0.474              | 0.267              | 0.201              |
| Solans | 0.862              | 0.332              | 0.262              |

Table 1: Average runtime per iteration for different attack types and datasets. All values are stated in units of seconds.

| Claim | Reproducible? |
|-------|---------------|
| *Claim 1* | Yes |
| *Claim 2* | Yes |
| *Claim 3* | No  |
| *Claim 4* | Yes |
| *Claim 5* | No  |

Table 2: Summary of the claims investigation under our specific setup.

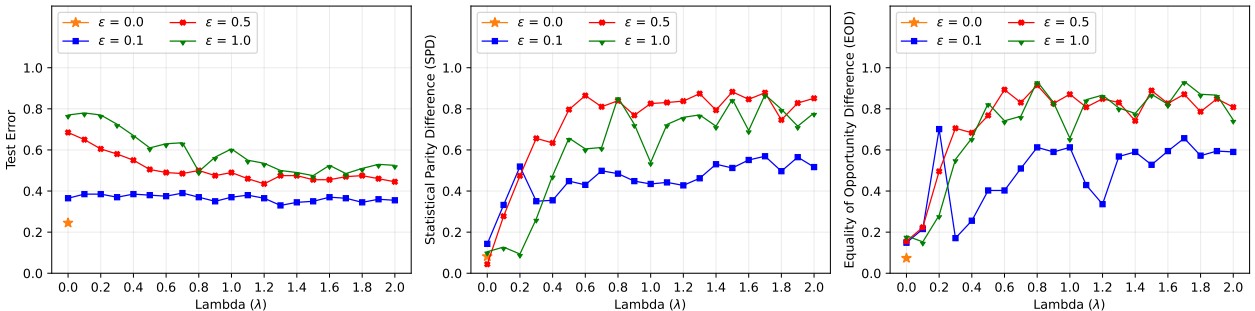

Figure 1: Influence of $\lambda$ on the different metrics for different $\epsilon$ on the German dataset, using accuracy as the stopping criteria during training.

section 4.1.1 we show the effect of the hyperparameter $\lambda$ on various metrics (*Claim 1*). In section 4.1.2 we compare the newly proposed attacks and the baselines (*Claims 2-5*).

### 4.1.1 Effect of $\lambda$ on the different metrics

To verify *Claim 1*, we conducted the same experiment as the authors. We run an IAF attack for each dataset using different $\epsilon$ values and increasing $\lambda$, to recreate Figure 3 of the original paper (see Appendix B.3, Fig. 8). However, compared to the original experiment we test a larger range of $\lambda$ values (from 0.0 to 2.0) to gain better insights into its effects. As depicted in Figure 1, increasing $\lambda$ does result in stronger attacks against fairness. Here we use the German dataset and accuracy as the stopping metric, but similar trends were observed on the other datasets and using fairness for early stopping. The plots are included in Appendix B.1 for the sake of completeness. Therefore in this specific setup, we were able to reproduce the claim.

### 4.1.2 Comparison between the proposed attacks and the baselines

To investigate *Claims 2-5* we design an experiment that is heavily inspired by the work of the authors. We perform each attack on each dataset, fixing $\lambda = 1$ and gradually increasing $\epsilon$ (from 0.0 to 1.0, with steps of 0.1), and repeat this procedure for each stopping metric. The results essentially replicate Figure 2 of the original paper (as seen in Appendix B.3, Fig. 7) and are collected in Figures 5 and 6 of Appendix B.2. However, to facilitate a comparative study between the proposed attacks and the baselines, we average the metrics over the $\epsilon$ values and report the results in Table 3. In this way, we can base our observations on quantifiable measures instead of solely using visual inspection.

Assuming that the authors used *accuracy* as the early stopping criteria, the corresponding values in the table reveal that – in this specific setting:

- *Claim 2* is reproducible. On average, IAF has a much stronger influence on SPD and EOD compared to Koh's attack, on all three datasets.
- *Claim 3* is not reproducible, because Solan's attack outperformed IAF in affecting the EOD on the Compas dataset.
- *Claim 4* is reproducible. NRAA and RAA were found to degrade the fairness metrics (SPD and EOD) more than Koh's attack, on all three datasets.
- *Claim 5* is not reproducible. Solans' attack had a greater impact on the SPD than NRAA on the German and greater impact than NRAA on both SPD and EOD on the Compas dataset. It also has a greater impact on the EOD than the RAA attack on the Compas dataset.

| | German Dataset | | | Compas Dataset | | | Drug Dataset | | |
|---|---|---|---|---|---|---|---|---|---|
| **Attack** | **Test error** | **SPD** | **EOD** | **Test Error** | **SPD** | **EOD** | **Test error** | **SPD** | **EOD** |
| | *(Stopping metric: Fairness / Accuracy)* | | | *(Stopping metric: Fairness / Accuracy)* | | | *(Stopping metric: Fairness / Accuracy)* | | |
| **IAF** | 0.40/**0.47** | **0.84**/0.68 | **0.88**/0.74 | 0.46/**0.47** | **0.83**/0.75 | **0.87**/0.77 | 0.43/**0.45** | **0.89**/0.75 | **0.90**/0.76 |
| **NRAA** | 0.26/0.26 | **0.26**/0.25 | **0.36**/0.33 | 0.41/**0.42** | 0.59/0.59 | 0.64/0.64 | 0.39/0.39 | 0.53/0.53 | 0.53/0.53 |
| **RAA** | 0.27/**0.28** | **0.24**/0.17 | **0.36**/0.19 | 0.47/0.47 | **0.84**/0.73 | **0.87**/0.75 | 0.42/**0.44** | **0.66**/0.55 | **0.68**/0.57 |
| **Koh** | 0.27/**0.61** | **0.17**/0.08 | **0.13**/0.12 | 0.45/**0.53** | **0.81**/0.46 | **0.85**/0.48 | 0.40/**0.56** | **0.56**/0.26 | **0.56**/0.29 |
| **Solans** | 0.40/**0.48** | **0.65**/0.44 | **0.49**/0.16 | 0.44/**0.45** | **0.76**/0.73 | **0.83**/0.78 | 0.40/**0.56** | **0.53**/0.28 | **0.55**/0.32 |

Table 3: Average metrics over $\epsilon$ values, obtained for each measure-attack combination and each dataset. We report one pair of values in each entry, corresponding to the two stopping criteria (average fairness and accuracy), and highlight the greatest one.

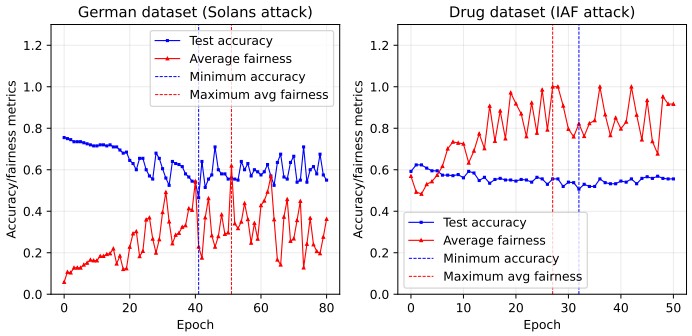

Figure 2: Difference between the two stopping metrics (accuracy and average fairness) for the Solans attack on the German dataset (left), IAF attack on the Drug dataset (right).

| **Value** | **German (Solans)** | **Drug (IAF)** |
|---|---|---|
| Min. test accuracy | 0.465 | 0.506 |
| Avg. fairness at the point of min. accuracy | 0.229 | 0.822 |
| Actual max. average fairness | 0.619 | 1.000 |

Table 4: Minimum accuracy, the value of the average fairness at the point of minimum accuracy, and maximum achievable average fairness of the plots of Figure 2.

### 4.2 Results beyond the original paper: using fairness as the early stopping metric

While the original codebase seems to use *accuracy* as the early stopping metric (and hence for selecting and saving the best model), we investigate the change in the results if *fairness* is used instead. The main motivation behind such an experiment lies in the assumption that interrupting training based on the fairness measures supposedly yields more relevant states of the poisoned data, effectively resulting in more efficient attacks against fairness. Since the SPD and EOD have similar behavior and equal range $[0, 1]$, we employ *average* fairness $(SPD + EOD)/2$ for the task at hand.

Figure 2 depicts the test accuracy and the average fairness over epochs for two different dataset-attack combinations. An analysis of the curves confirms that the maximum achievable average fairness is much greater than the same measure at the point of minimal accuracy (see Table 4). The same phenomenon is observed for *any* dataset-attack combinations, as reported in Table 3: fairness undergoes a stronger degradation if average fairness is used to interrupt the training process and save the best model. This is reflected in the corresponding values of the fairness measures, which appear much higher compared to when accuracy is used.

## 5 Discussion

Our reproduction reveals that although the proposed methods represent valid novel attacks against the fairness of a model, they are not always superior to other methods in the literature. IAF showed important performance in terms of SPD and EOD degradation, but anchoring attacks were outperformed by the baseline models on multiple occasions. This result conflicts with the findings of the main paper (see Appendix B.3, Fig.7) where the baselines are generally inferior to the proposed attacks. We had to make several assumptions to solve issues and inconsistencies between the original paper and corresponding implementation (many of which have already been mentioned throughout the report, but we systematically collect them in Appendix A). These assumptions are, by definition, uncertain and might have been the cause of the discrepant results. To better understand the source of discrepancy, we initially planned to perform an ablation study, which would have also unveiled more information regarding the model's behavior. This was ultimately not possible, given the time constraints and the contingencies encountered in the reproduction process.

In the remainder of this section, we elaborate on the main claims and our ability to reproduce them. We then present some personal reflections on the overall execution of the work and conclude with a summary and look into future works.

## 5.1 Discussion of the results

The first claim was found to be reproducible under our experimental setup, as we expected. The parameter $\lambda$ is specifically designed to control the trade-off between accuracy and fairness, hence a rejection of the claim would have implied a major flaw in the core idea of the paper. The other claims focused on the comparison with the two baselines and, while the results presented in Section 4.1.2 are explicative enough, some remarks are still noteworthy.

In general, better statistics of the results would give us a clearer insight into the relative performance of the models. However, only four weeks were allocated for this project and we were unable to re-run the experiments with multiple seeds. For example, the Solans attack outperformed the IAF attack in terms of EOD metric on the Compas dataset (when using accuracy as the stopping method) and led to the non-reproducibility of *Claim 3*. Yet, this difference is relatively small and a measure of uncertainty could potentially reverse our decision.

Furthermore, it was shown that the final fairness metrics can highly vary depending on the chosen stopping method. This is especially prominent for *Claim 4*, which was accepted under the assumption that accuracy was used for stopping and saving the best model. In reality, Koh attack outperforms NRAA on both Compas and Drug datasets in the terms of SPD/EOD metrics, if fairness is used instead. Since the validity of the claim depends on the stopping metric of choice, we argue that the claim is much weaker than originally proposed. Similarly, compare the IAF and the Koh attack in terms of fairness measures, using accuracy as the stopping criteria. On the Drug dataset, IAF's SPD/EOD metrics are respectively $2.89\times/2.62\times$ higher than Koh's. This gap tightens if fairness is used: IAF's SPD/EOD metrics become $1.022\times/1.024\times$ higher. Although these numbers indicate the same result, we find the claim to be weaker than proposed, as the superior performance of the IAF attack is diminished by the use of a different stopping metric.

Finally it is important to notice the different behavior of the test accuracy and the average fairness (Fig. 2) used as stopping criteria. While the latter has a relatively high variance, the former is pretty constant, meaning that using fairness as the stopping metric does not result in significant variations in the model's accuracy. Contrarily, as empirically proved by our experiments, it can be highly beneficial for the fairness measures.

## 5.2 Reflection: What was easy? What was difficult?

The new methods presented in the paper were described both intuitively and formally, with a clear mathematical structure. The authors also provided figures to aid the intuition on how new attacks can affect decision boundaries, which allowed us to easily understand the core novel ideas presented in the publication.

However, it was not trivial to re-implement the proposed methods, because many details required for the implementation do not appear in the paper. The provided open-source implementation was ultimately hard to follow due to its convoluted organization, lack of documentation, poorly named functions/variables, and abundance of unused code. Even setting up a working environment using the authors-given dependencies took longer than one would expect, prompting us to get help from the authors. Eventually, the hope to aid future experiments motivated the decision to make the code compatible with up-to-date dependencies. This was one of the biggest struggles because the codebase heavily relies on packages that underwent major updates (e.g. `TensorFlow` and `CVXpy`).

The authors also provided pre-processed datasets. We spent a considerable amount of time trying to replicate their exact pipeline through reverse-engineering of the given files. Additionally, after recognizing some imperfections in the code and inconsistencies with the paper, we verified all of the existing implementation details to make sure that no further errors were made. This was a daunting task, given the complete lack of documentation and intuitive variable use.

## 5.3 Communication with original authors

To reiterate, we have initially contacted the main author to aid us with the dependency issues, who helped us with setting up a working environment. We then had additional contacts regarding the dataset pre-processing procedure. The author provided us with some indications on the pipeline and pointed at some useful resources. Eventually, we decided to gain a better understanding of the datasets through reverse-engineering.

## 5.4 Conclusion

In this paper, we have presented a reproducibility study of "Exacerbating Algorithmic Bias through Fairness Attacks", whereon we can draw some conclusions. Due to all the mentioned issues and inconsistencies (collected in Appendix A), we find it not possible to reproduce the original results from sole use of the paper, and difficult even in possession of the provided codebase. Yet, we managed to obtain similar findings that supported three out of the five main claims of the publication, albeit using partial re-implementations and numerous assumptions. Ascertaining the validity of such assumptions is therefore important for future works. Moreover, further studies could extend the classifier to work with multiple demographic groups and investigate the results using different fairness metrics.

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

# A   Table of issues

| Issue | Our contribution |
| --- | --- |
| Running the code in the given environment results in conflicts | Code modernized and made compatible with the latest version of every dependency |
| The code is generally complex and hard to understand due to insufficient comments and documentation as well as leftover code | Trimmed down the code to the essential, included option to choose any of the available models and the corresponding parameters. Added comprehensive documentation to make the code more interpretable |
| It appears that the pre-processing pipeline of the given datasets is not specified | Made the scripts we used to pre-process each dataset available as well as a detailed description |
| It appears that the position (i.e. index) of the sensitive feature for the COMPAS and Drug Consumption datasets is not indicated, posing a challenge to reproduce the authors' results | Moved the sensitive feature (i.e. gender) of every dataset to the 0th index, which is taken as default by the main function |
| The advantaged and disadvantaged groups for the sensitive attribute (gender) has not be specified for any attack | Assumed from the codebase that the authors did this automatically and inferred it from the dataset (the advantaged group is chosen as the group with a higher ratio of datapoints with the positive label (y=1), regardless of the actual class label it corresponds to) to be specified for all attacks |
| The code does not provide an option to run baseline methods used in the paper, nor does it include the hyperparameter $\lambda$ | Included option to run baseline methods (Koh attack, Solans attack) and to include $\lambda$ in IAF attack |
| The code implements a deterministic point sampling in the anchoring attacks (RAA, NRAA) due to the same seed being reset in every attack iteration. Thus the sampling yields the same point every iteration not properly applying the randomness | Fixed the issue so that randomness takes effect |
| The code makes use of a faulty EOD metric calculation | Re-implemented the EOD metric calculation to fix the issue |
| The paper specifies the feasible set computation to be done on the union of the clean dataset and the initial poisoned points. The original code however does this on the clean data only when using the running commands given by the authors | Implemented the feasible set as specified in the paper |
| It appears that the model used for the given classification task is not specified | Assumed they used a Support Vector Machine (SVM) trained with a smooth hinge loss and L2 regularization |
| The optimization algorithm is not indicated | Assumed it to be Newtons Conjugate Gradient (Newton-CG) method, as suggested by the codebase |
| It is unclear how the best performing model was selected | Saved best performing model on the test set according to the chosen stopping metric |

# B   Additional figures

Here we collect additional figures that support the results discussed above.

## B.1   Effect of $\lambda$ on the different metrics

Figure 3 shows the influence of $\lambda$ on the different metrics when *accuracy* is used as the stopping criteria. The experiment is repeated using *average fairness* as the stopping metric, and the results are collected in Figure 4. These results support *Claim 1* of Section 2, effectively proving it.

## B.2  Comparative study between the proposed attacks and the baselines

We report the results of the experiment designed to support *Claims 2-5* of Section 2. We perform each attack (IAF,
NRAA, RAA, Koh, Solans) on each dataset (German, Compas, Drug), fixing $\lambda = 1$ and gradually increasing $\epsilon$ from 0.0
to 1.0, with steps of 0.1. We repeat this procedure for each stopping metric (average fairness and accuracy). The results
are respectively collected in Figures 4 and 5.

## B.3  Figures of the original paper

For the sake of self-containedness of this reproducibility study, we report the two main figures of the original paper.
Figures 7 and 8 correspond – respectively – to Figures 2 and 3 of "Exacerbating Algorithmic Bias through Fairness
Attacks".

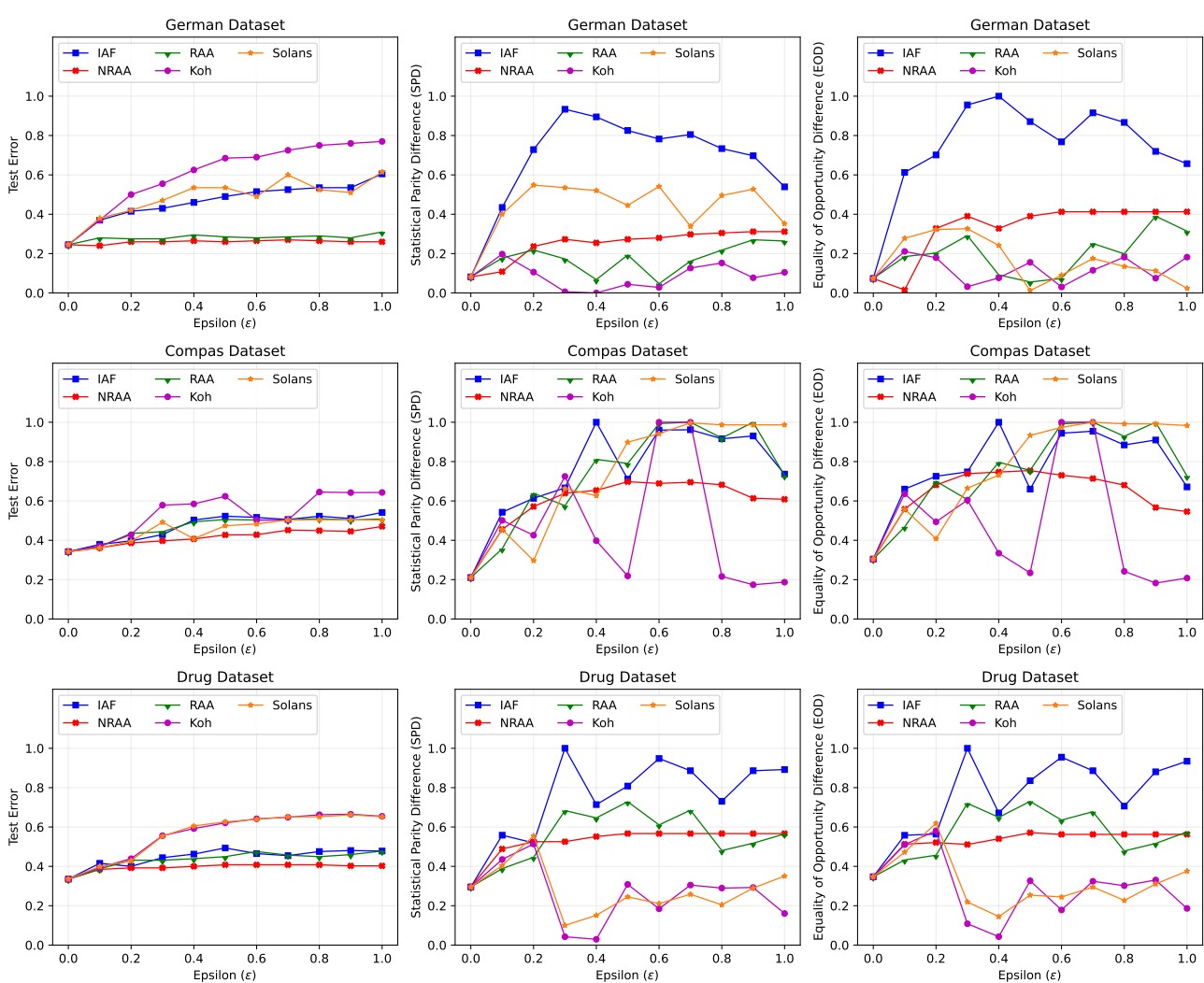

Figure 3: Results obtained for different attacks with regards to accuracy (test error) and fairness (SPD and EOD)
measures on German Credit, COMPAS, and Drug Consumption databases with different $\epsilon$ values and with *accuracy* as
the stopping method.

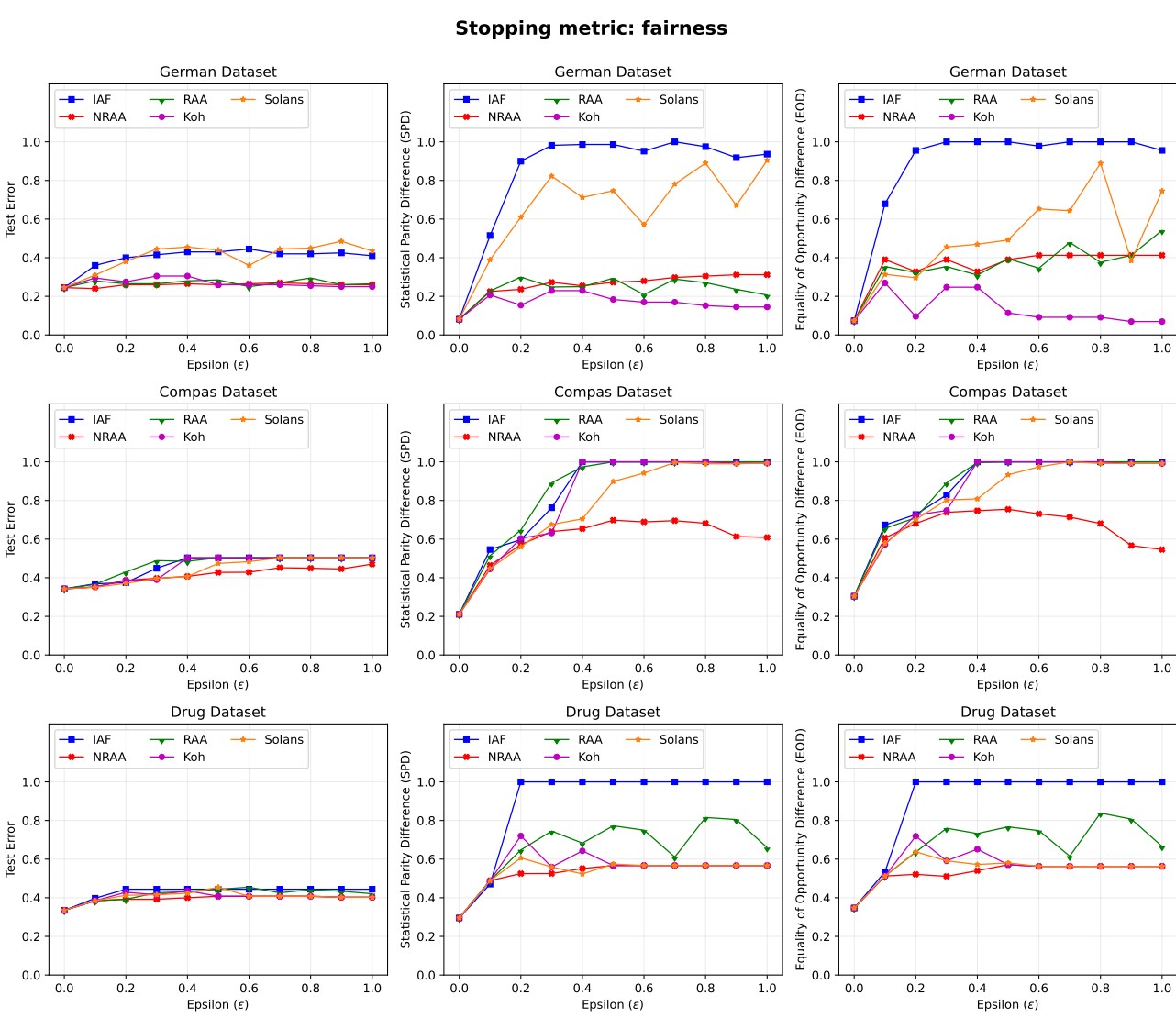

Figure 4: Results obtained for different attacks with regards to accuracy (test error) and fairness (SPD and EOD) measures on German Credit, COMPAS, and Drug Consumption databases with different $\epsilon$ values and with *average fairness* as the stopping method.

**Stopping metric: accuracy**

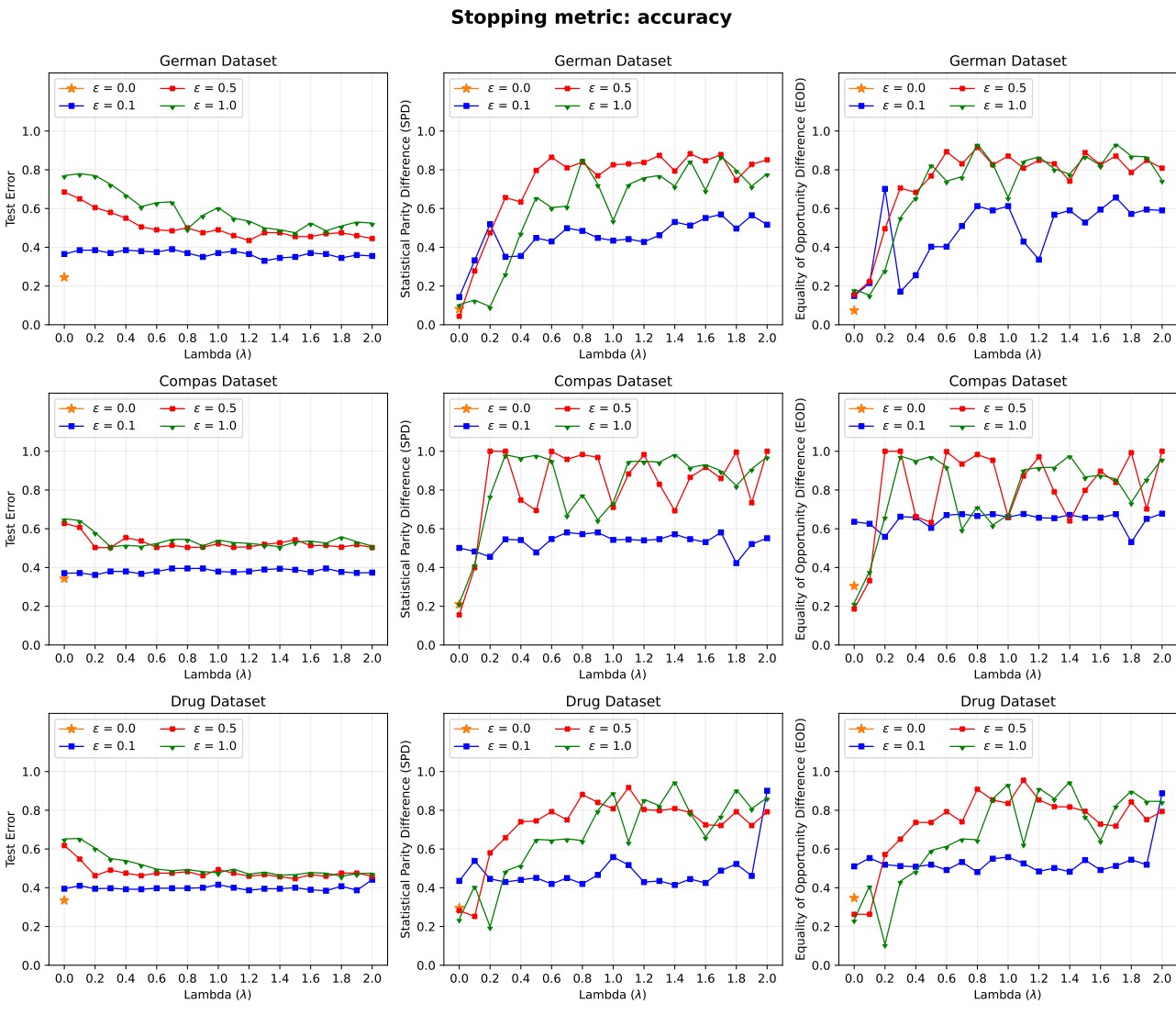

Figure 5: Accuracy (test error) and fairness (SPD and EOD) measures obtained after the IAF attack the on German Credit, COMPAS, and Drug Consumption databases for different $\epsilon$ and increasing $\lambda$ values, with *accuracy* as the stopping method.

**Stopping metric: fairness**

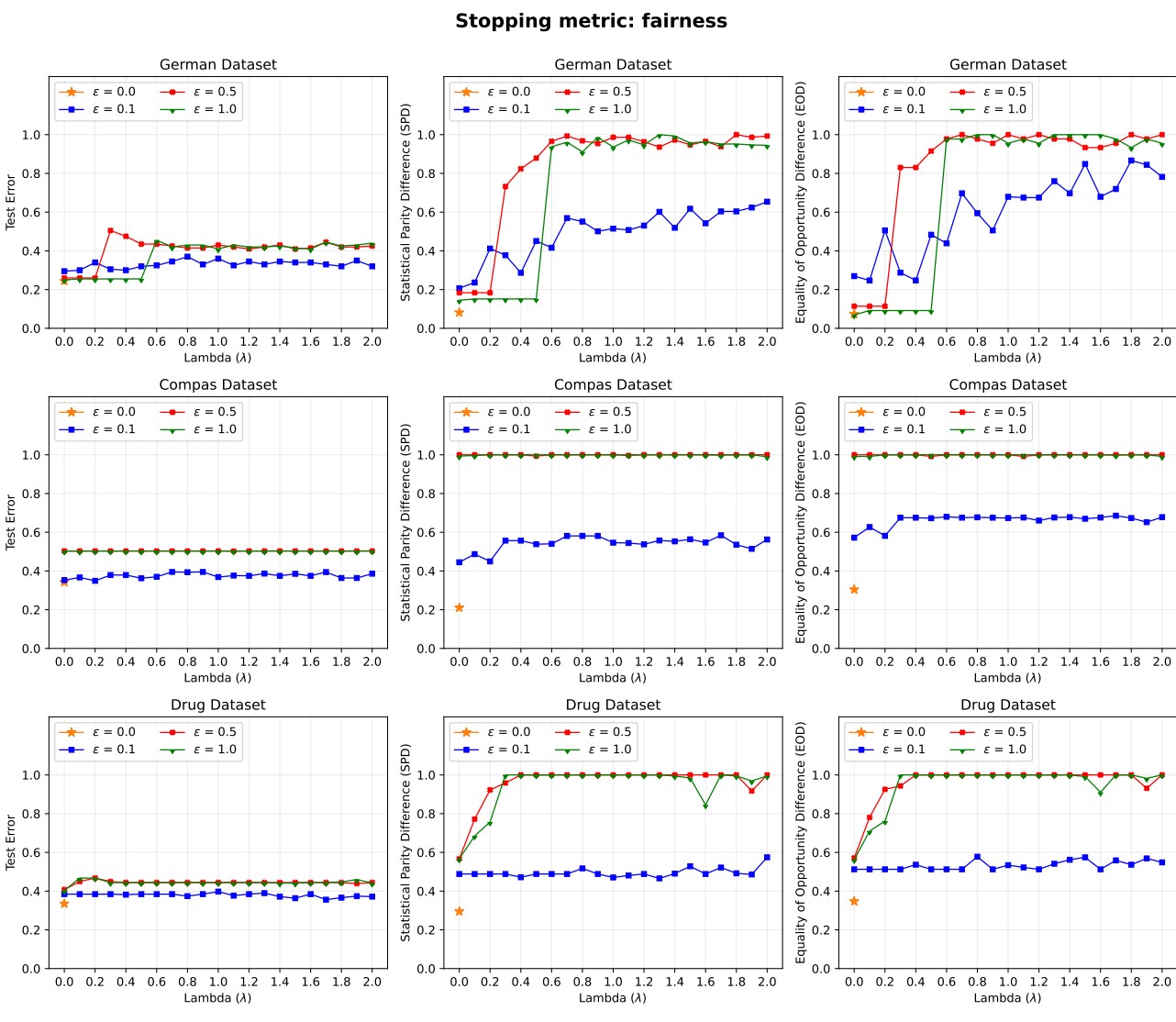

Figure 6: Accuracy (test error) and fairness (SPD and EOD) measures obtained after the IAF attack the on German Credit, COMPAS, and Drug Consumption databases for different $\epsilon$ and increasing $\lambda$ values, with *average fairness* as the stopping method.

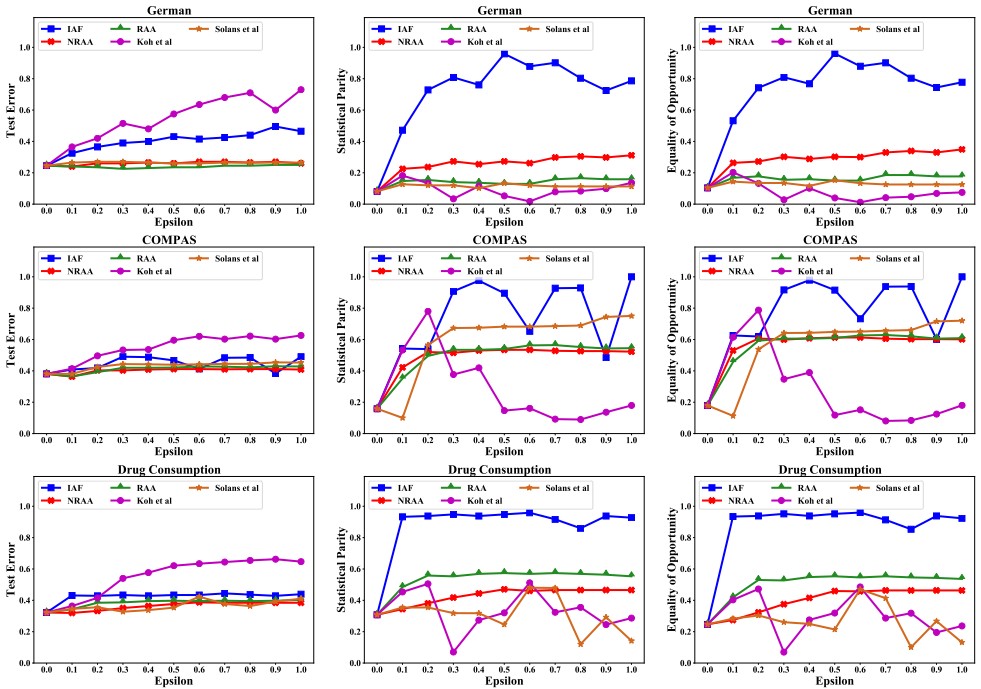

Figure 7: Results of the original paper obtained for different attacks with regards to different fairness (SPD and EOD) and accuracy (test error) measures on three different datasets (German Credit, COMPAS, and Drug Consumption) with different $\epsilon$ values. Retrieved from [1].

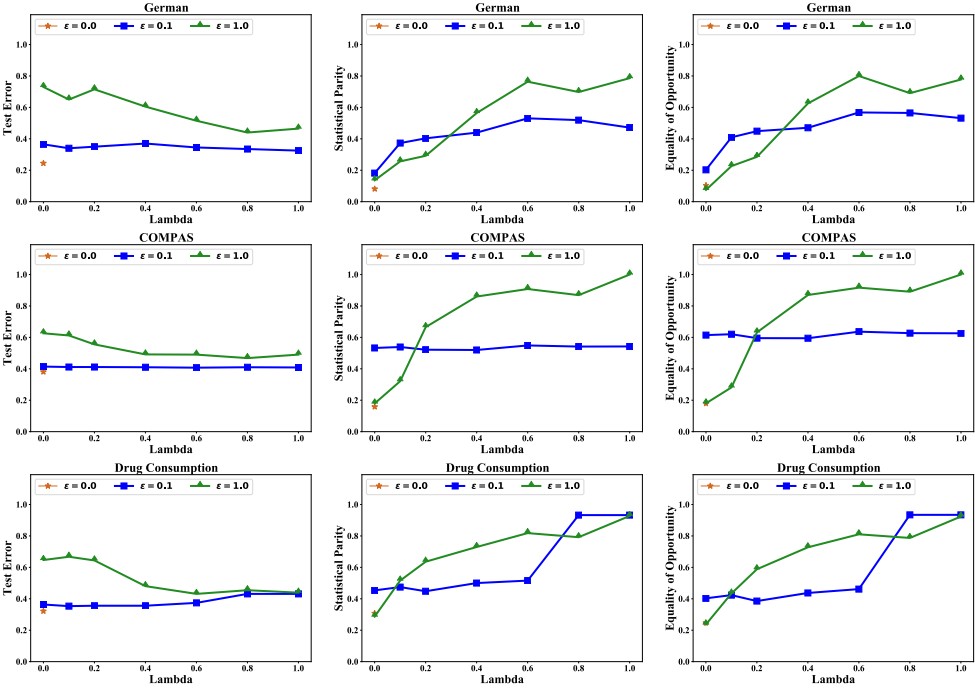

Figure 8: Results of the original paper obtained for different $\lambda$ values for the IAF attack with regards to different fairness (SPD and EOD) and accuracy (test error) measures on three different datasets (German Credit, COMPAS, and Drug Consumption) with different $\epsilon$. Retrieved from [1].

