# OpenReview forum: "On the reproducibility of "Exacerbating Algorithmic Bias through Fairness Attacks""
_ML_Reproducibility_Challenge/2021/Fall — RC2021 OutstandingPaper_

### Official Review · Reviewer_jrT1 · 2022-02-26
**Clear report on reproducibility of fairness attacks**

**Rating:** 9
**Confidence:** 4

**Review:**

This submission investigates the reproducibility of a recent work on attacks targeted at fairness in algorithms. The authors assess the reproducibility of five main claims in the original work, and their analysis supports three of these claims. The authors also extend the original work by implementing new baseline attacks, for added robustness analysis, and suggesting a modification to influence the performance of certain types of attacks.

Quality & clarity: The report very clearly and concisely states the results and the scope of reproducibility, and adheres to it. Technical content is balanced by clear structure and discussion. types of attachs and main claims to explore are stated clearly. Concrete shortcomings in the original reporting and code are pinpointed, and this provides good guidance for improving reproducibility in computational analyses and reporting, including supporting visualizations and a summary table of the issues and newly implemented solutions. The authors have done notable additional work to improve the executability and documentation by modifying the original code base.

Originality & significance: Communication with the original authors is reported, and it has been relevant for reproducing the results. This work provides partial support to the claims in the original work but also identifies many shortcomings in the reporting and analysis reproducibility. Detailed reproducibility analysis steps are included and it is indicated that the original report and source code do not seem to be sufficient for a full replication.

Code availability: Codebase is available via an anonymized Github repository and seems clearly documented.

Pros:
- The reporting is sufficiently comprehensive and easy to follow despite the inevitable technicality of the content, and it includes useful remarks and recommendations that support independent reproducibility analyses; good use of text structuring, illustrations, and summary table help to follow the text
- Notable extra work to improve the executability and documentation on top of the original code base
- Additional new contributions include two new baseline attacks and a modification that can influence efficiency of certain types of attacks
- Source code of the reproducibility analysis is available


Cons:
- Is the LICENSE copyright information is up-to-date? Source code of the reproducibility analysis has an open license but the LICENSE file names the author of the original work as the copyright holder; but according to the authors of the reproducibility report, the original code has been remarkably augmented.
- Table 2 makes somewhat strong claims about the non-reproducibility of the results; strong support was not identified and there is limited reproducibility (no reproducibility is a stronger claim and concerns a specific, limited setup). This statement could be mitigated to also acknowledge the limitations of the reproducibility analysis itself.

---

### Official Review · Reviewer_ZKbN · 2022-03-19
**Review on: On the reproducibility of "Exacerbating Algorithmic Bias through Fairness Attacks"**

**Rating:** 9
**Confidence:** 5

**Review:**

- Reproducibility Summary: Present
- Scope of reproducibility: Clearly Stated
- Code: re-used author repository
- Communication with original authors: yes
- Hyperparameter Search: yes; includes new hyperparameters not tried by the authors
- Ablation Study: Comprehensive
- Discussion on results: Extensive discussion; detailed description of the reproducible/non-reproducible parts
- Recommendations for reproducibility: Useful criticism for the authors
- Results beyond the paper: Yes;
- Overall organization and clarity: Good

---

### Meta-Review · Area_Chair_yLSX · 2022-04-07

**Recommendation:** Accept (Outstanding Paper)
**Confidence:** 5

**Metareview:**

The paper has very strong reviews. I agree with the reviewers that the paper has an exceptional quality in reproduction of the original results and clarity in presentation. I recommend for its acceptance.

---

### Decision · Program_Chairs · 2022-04-09

**Decision:**

Accept (Outstanding Paper)

**Comment:**

Following the recommendation of reviewers and meta-reviewer, the paper is accepted for ML Reproducibility Challenge 2021, and will be published in the upcoming special edition of ReScience Journal.

Additionally, after several rounds of discussion and incorporating recommendations from the Area Chairs and Program Chairs, the report has been granted an **Outstanding Paper Award** due to its exceptional quality of all-round reproducibility effort. Congratulations!